# Cytoskeletal vimentin regulates cell size and autophagy through mTORC1 signaling

**Ponnuswamy Mohanasundaram**[1,2☯], **Leila S. Coelho-Rato**[1,2☯], **Mayank Kumar Modi**[1,2], **Marta Urbanska**[3,4¤], **Franziska Lautenschläger**[5,6], **Fang Cheng**[1,2,7], **John E. Eriksson**[1,2]*

**1** Turku Bioscience Centre, University of Turku and Åbo Akademi University, Turku, Finland, **2** Cell Biology, Faculty of Science and Engineering, Åbo Akademi University, Turku, Finland, **3** Biotechnology Center, Center for Molecular and Cellular Bioengineering, Technische Universität Dresden, Dresden, Germany, **4** Max Planck Institute for the Science of Light & Max-Planck-Zentrum für Physik und Medizin, Erlangen, Germany, **5** Saarland University, NT Faculty, Experimental Physics, Saarbrücken, Germany, **6** Center for Biophysics, Saarland University, Germany, **7** School of Pharmaceutical Sciences (Shenzhen), Shenzhen Campus of Sun Yat-sen University, Shenzhen, Guangdong, P.R. China

☯ These authors contributed equally to this work.
¤ Current address: Department of Physiology, Development and Neuroscience, University of Cambridge, Cambridge, United Kingdom
* john.eriksson@bioscience.fi

**Data Availability Statement:** The authors confirm that all data underlying the findings are fully available without restriction. Flow cytometry files are available in the Flowrepository database https://

## Abstract

The nutrient-activated mTORC1 (mechanistic target of rapamycin kinase complex 1) signaling pathway determines cell size by controlling mRNA translation, ribosome biogenesis, protein synthesis, and autophagy. Here, we show that vimentin, a cytoskeletal intermediate filament protein that we have known to be important for wound healing and cancer progression, determines cell size through mTORC1 signaling, an effect that is also manifested at the organism level in mice. This vimentin-mediated regulation is manifested at all levels of mTOR downstream target activation and protein synthesis. We found that vimentin maintains normal cell size by supporting mTORC1 translocation and activation by regulating the activity of amino acid sensing Rag GTPase. We also show that vimentin inhibits the autophagic flux in the absence of growth factors and/or critical nutrients, demonstrating growth factor-independent inhibition of autophagy at the level of mTORC1. Our findings establish that vimentin couples cell size and autophagy through modulating Rag GTPase activity of the mTORC1 signaling pathway.

## Introduction

Cell size regulation is intricately related to nutrient availability and the rate by which macromolecules are synthesized [1]. To maintain stable cell growth, cells require nutrients and growth factors. On the other hand, when cells are deprived from nutrients or growth factors, or if they are exposed to stress, cell growth will be inhibited and autophagy will be triggered in order to recycle cellular components to ensure cell survival. These processes, both cell growth and autophagy, are primarily governed by the mechanistic target of rapamycin kinase complex 1 (mTORC1) signaling complex. mTORC1 acts as central signaling hub in regulating cell size

flowrepository.org/id/FR-FCM-Z5FK. The gating strategy used to analyze the cell populations is deposited in Figshare (https://doi.org/10.6084/m9.figshare.20024534.v1). All other relevant data are within the paper and its Supporting Information files.

**Funding:** This study was supported by following funders below: Academy of Finland #317867 (https://www.aka.fi/en/) to JEE, Sigrid Jusélius Foundation (https://www.sigridjuselius.fi/en/) to JEE, Magnus Ehrnrooth Foundation (https://www.magnusehrnroothinsaatio.fi/en/) to PM, LSCR, The Endowment of the Åbo Akademi University (https://stiftelsenabo.fi/) to JEE, K. Albin Johanssons stiftelse (https://www.foundationweb.net/johansson/) to PM, LSCR, Maud Kuistila Memorial Foundation (https://mkmsaatio.fi/en/the-maud-kuistila-memorial-foundation/) to LSCR, Liv och Hälsa Foundation (http://www.livochhalsa.fi/) to PM, Otto A Malm Foundation (https://en.ottomalm.fi/) to LSCR, Finnish Cultural Foundation (https://skr.fi/en) to LSCR, Swedish Cultural Foundation (https://www.kulturfonden.fi/in-english/) to LSCR, Ella and Georg Ehrnrooth Foundation (https://www.ellageorg.fi/en) to LSCR, The Foundation "Konung Gustaf V:s och Drottning Victorias Frimurarestiftelse" (https://www.kungahuset.se/kungliga-stiftelser/forskning) to JEE, The DFG German Research Foundation # CRC 1027(https://www.dfg.de/en/) to FL. The funders had no role in study design, data collection and analysis, decision to publish, or preparation of the manuscript.

**Competing interests:** The authors have declared that no competing interests exist.

**Abbreviations:** BMDCs, bone marrow-derived dendritic cells; EAAs, essential amino acids; EMT, epithelial to mesenchymal transition; FCS, fetal calf serum; IFs, intermediate filaments; mTORC1, mechanistic target of rapamycin kinase complex 1; NEAAs, non-essential amino acids; PFA, paraformaldehyde; RT-DC, real-time deformability cytometry; SFM, serum-free media.

and metabolism by sensing and coordinating stimuli derived from nutrients, energy, stress, and growth factors [2]. Amino acids mediate mTORC1 signaling through Rag GTPases: Presence of amino acids activate Rag GTPases leading to recruitment of mTORC1 to the lysosomal membrane where it is activated by Rheb [3–5]. This allows mTORC1 to promote protein synthesis by controlling downstream effectors, 4-EBP1 and p70S6K [6] and by inhibiting autophagy via phosphorylation of ULK1, a known promoter of autophagy [7–9].

Vimentin, an intermediate filaments (IFs) protein, has been reported to act as signaling scaffold in key cellular processes required to maintain tissue integrity and facilitate tissue repair [10,11]. These include cell migration, adhesion, proliferation, and invasion [12]. In this respect, vimentin is a well-established marker for epithelial to mesenchymal transition (EMT), a crucial step in wound healing and metastasis [13]. Vimentin-deficient (Vim −/−) mice exhibit delayed wound healing due to defects in EMT signaling, cell migration, and cell proliferation [14]. Furthermore, vimentin regulates autophagy by forming a complex with the autophagy regulator Beclin and the adaptor protein 14-3-3 [15]. Another link between cell size and vimentin was provided by a recent study showing that Vim −/− mice have deficient accumulation of body fat [16] and by the first report on a human vimentin mutation leading to lipodystrophy [17]. Keratin 17, another member of the IF family, was implicated in cell size signaling, as mouse-derived skin keratinocytes lacking keratin 17 are smaller than corresponding WT cells and display lower AKT/mTOR activation [18]. Related to all these studies, we report here that cell size regulation is coupled to vimentin, both in terms of enlargement and reduction.

## Results and discussion

### Loss of vimentin reduces cell size

When reporting that vimentin has a role in fibroblast proliferation and in EMT [14,19], we observed that not only do Vim −/− fibroblasts grow slower, but also they were significantly smaller (S1 Video). Correspondingly, when examining the Vim −/− mice in greater detail, we observed that they were leaner, as reflected in remarkable reductions in weight, body mass index, as well as Lee's index (Fig 1A–1C), in line with an initial recent report [16]. These mice also had markedly lower fat content (Fig 1D) and smaller size of adipocytes (Fig 1E). Likewise, some of the organs (heart and kidney) were also smaller in size (Fig 1F). As all these effects point toward a potential disturbance in cell size regulation, we quantified in detail the effects of vimentin on cell size. We found that Vim −/− MEFs are significantly smaller than WT MEFs (Fig 2A and 2B). This effect was seen in cells in their active growth phase and cells in confluent state (S1 Video), indicating that the effect is not coupled to cell spreading. To eliminate the effect of cell spreading, we next measured the cell volume of trypsinized MEFs and found that the volume of Vim −/− MEFs is considerably lower than that of WT MEFs (Fig 2C and 2D). To minimize cell cycle-dependent size variation, we employed thymidine-induced cell cycle arrest, which revealed an even more pronounced cell volume reduction in the Vim −/− MEFs as compared to the WT MEFs, demonstrating that the size effect on cell volume is cell cycle independent (Fig 2E). To ensure that these effects were not specific for the immortalized MEFs, we assessed the size of primary MEFs isolated from WT and Vim −/− mice, which showed a similar size reduction in the Vim −/− primary MEFs (Fig 2F). We also investigated the size of primary bone marrow-derived dendritic cells (BMDCs) using a high-throughput microfluidics-based method [20] to compare the results acquired from adherent fibroblasts with a non-adherent and spherical cell type. These cells possess a less extensive cytoskeleton, which implies that the possibility of mere size effects from removing vimentin as a cytoskeletal element is minimized. Furthermore, we used a

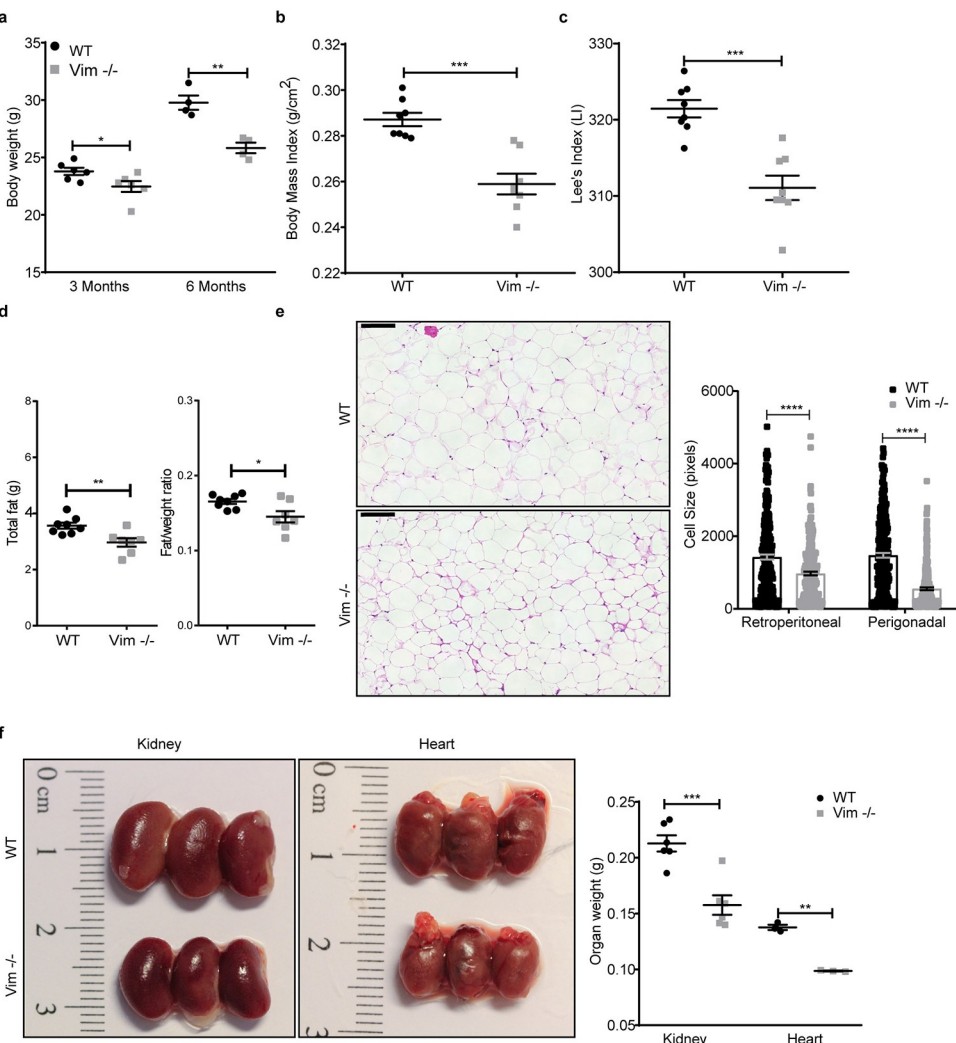

**Fig 1. Vim −/− mice are smaller than WT.** (a) Body weight of WT and Vim −/− male mice of 3 ($n = 6$) and 6 months old ($n = 4$). (b) BMI and (c) Lee's indexes were calculated for 6 months old mice ($n = 8$). (d) Total fat mass determined by EcoMRI and ratio of fat/weight of 6 months old male mice ($n = 7$–8). (e) Micrograph of adipocytes from 3 months old WT and Vim −/− male mice. The size of retroperitoneal and perigonadal adipocytes were quantified using Fiji ($n = 3$, 250 cells in total). (f) Organ weight of kidney ($n = 6$) and heart ($n = 3$) from 3 months old male mice. The results are presented in the form of mean ± standard deviation of the mean of the biological replicates. Scale bar = 100 μm, $^*p < 0.05$; $^{**}p < 0.01$, $^{***}p < 0.001$, $^{****}p < 0.0001$. The data underlying the graphs shown in Fig 1A–1F can be found in S1 Data.

microfluidic method to image the cells not only in the initial state, but also upon deformation in a narrow constriction of a microfluidic channel, where cytoskeletal structures are strained. Consistently with the above results, BMDCs lacking vimentin showed decreased size in both normal and deformed state (S1a and S1b Fig). These results emphasize that the vimentin-dependent effect on cell size is a generic effect irrespective of cell type. Furthermore, these results do not support the assumption of the cell size reduction stemming merely from the loss of the cytoskeletal component. To demonstrate that the effects were due to vimentin, we also show that the reduced cell volume could be rescued by reintroducing WT vimentin into Vim −/− MEFs (Fig 2G and 2H).

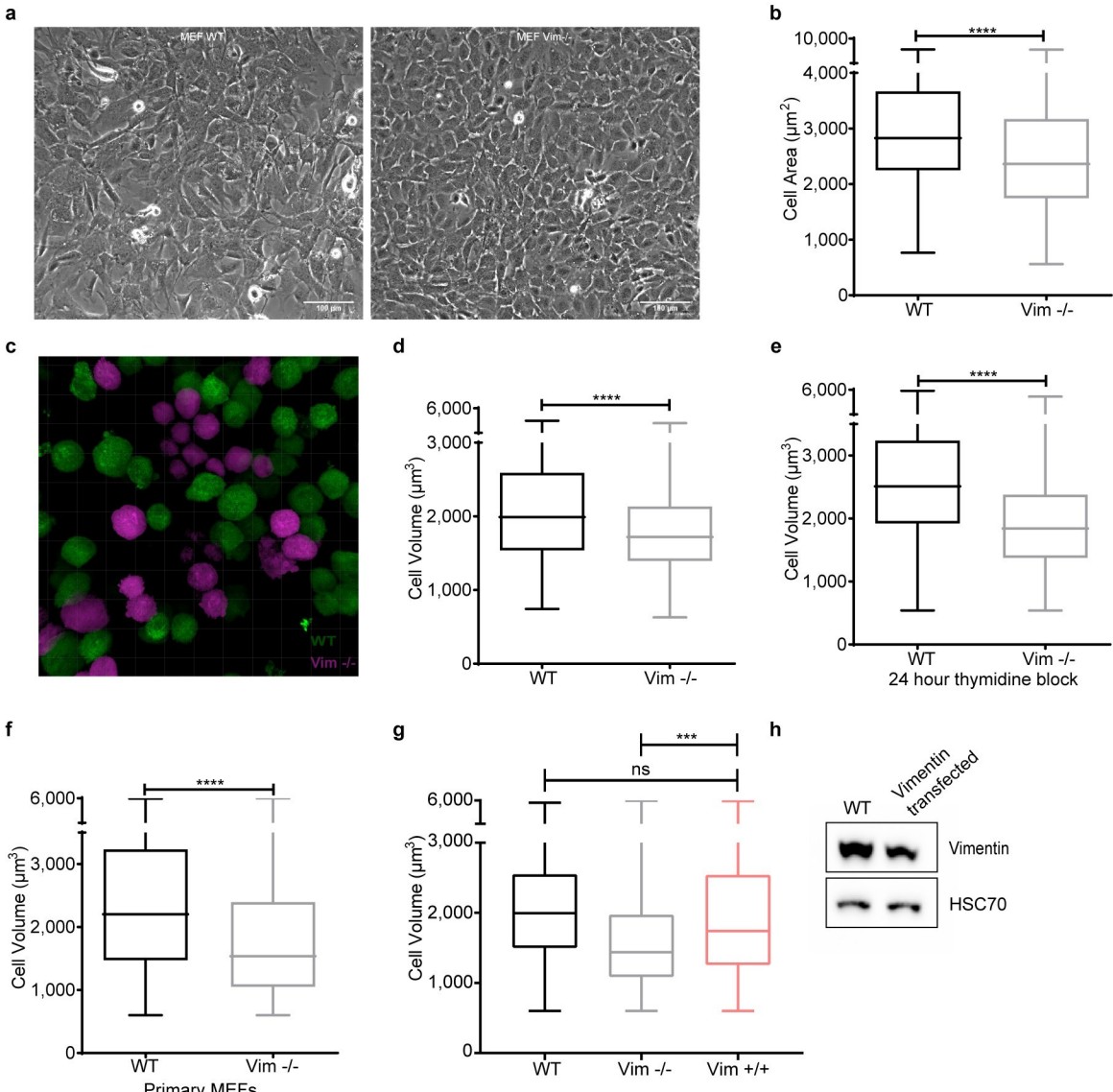

**Fig 2. Loss of vimentin reduces cell size.** (**a**) Phase contrast representative images of WT and Vim −/− MEFs (scale bar = 100 μm). (**b**) Distribution of the cell area obtained from phase contrast images measured using Fiji ($n$ = 3). (**c**) Micrograph of WT (green) and Vim −/− (magenta) MEFs image made with Imaris software for cell volume analysis. (**d**) Cell volume distribution of WT and Vim −/− MEFs obtained from z-stacks from cells in suspension stained with CellTracker ($n$ = 3). (**e**) Cell volume distribution of WT and Vim −/− cells treated with 1 mM of thymidine for 24 hours ($n$ = 3). (**f**) Cell volume distribution of primary MEFs isolated from WT and Vim −/− mice ($n$ = 3). (**g**) Cell volume distribution of WT, Vim −/−, and pCMV script-vimentin transfected (48 hours post transfection) in Vim −/− MEFs ($n$ = 3). (**h**) Western blot analysis of WT and Vim −/− MEFs transfected with wild-type vimentin. $^{***}p < 0.001$, $^{****}p < 0.0001$. The data underlying the graphs shown in Fig 2B and 2D–2G can be found in S1 Data.

## mTORC1 activation is impaired in Vim −/− MEFs

As cell size is tightly regulated by nutrients and growth factors [1], we wanted to assess whether the effect seen on cell size is linked to these stimuli. To this end, cells were serum starved for 2 to 3 days, then transferred to fresh serum-free media, and subsequently stimulated with insulin or fetal calf serum (FCS) for 24 hours. Importantly, we found that serum starvation reduced the size of WT MEFs to the same as Vim −/− MEFs (Fig 3A). Stimulation with insulin only, which rules out the influence of other growth factors, increased cell size only in WT MEFs and

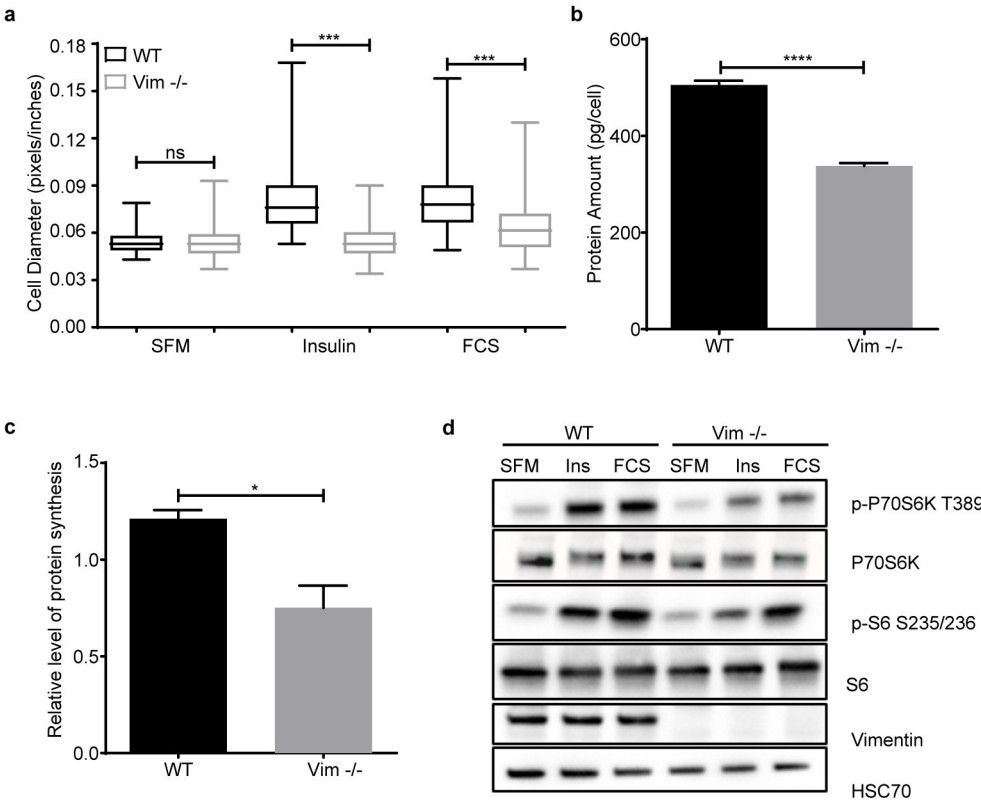

**Fig 3. mTORC1 activation is impaired in Vim −/− MEFs.** (**a**) Distribution of the cell diameter of WT and Vim −/− MEFs serum starved for 2 to 3 days and treated with 100 μg/ml insulin or 5% FCS for 24 hours (*n* = 3). (**b**) Protein amount per cell is represented as mean ± SEM of protein concentration. A same number of cells were lysed, and protein concentration was measured using a commercial BCA kit (*n* = 3). (**c**) Protein synthesis is represented as mean ± SEM of WT and Vim −/− MEFs stimulated with insulin for 30 minutes and it was measured by using the Click-iT HPG Alexa Fluor Protein Synthesis Assay Kit (*n* = 3). Protein synthesis was quantified using the mean fluorescence intensity normalized to the serum-starved treatment. (**d**) Western blotting analysis of serum-starved cells (overnight) treated with 100 nM of insulin or 5% serum for 15 minutes (*n* = 3). *$p < 0.05$, ***$p < 0.001$, ****$p < 0.0001$, ns = nonsignificant. The data underlying the graphs shown in Fig 3A–3C can be found in S1 Data. SFM, serum-free media; FCS, fetal calf serum; mTORC1, mechanistic target of rapamycin kinase complex 1.

not in Vim −/− MEFs (Fig 3A). Consistently, when the cells were stimulated with FCS, there was a significant increase in WT MEFs, whereas Vim −/− MEFs were unaffected (Fig 3A). Our results show that the reduction in cell size in Vim −/− MEFs is linked to a disruption in insulin-dependent signaling. Thus, the smaller size stems from abrogation of the cellular signaling mechanisms regulating this attribute and not from structural effects of a missing cytoskeletal component. Since cell size depends on the balance between synthesis and degradation of macromolecules [1], we measured protein amount per cell and found that it was significantly lower in Vim −/− MEFs (Fig 3B), indicating that the reduction in cell size is directly coupled to the anabolic state of the cells. To link this observation to growth signaling, we measured the level of protein synthesis after insulin stimulation and observed that it was significantly lower in Vim −/− MEFs (Fig 3C). These results demonstrate that vimentin participates in signaling stimulating cell size. Interestingly, starvation reduced the size of WT MEFs to the size of vimentin-deficient cells, while the size of the Vim −/− MEFs was not affected. This implies that the Vim −/− MEFs behave as if they would be in a compromised nutritional state already without starvation, due to deficient nutrient-coupled signaling.

It is well established that mTORC1 is one of the main pathways regulating cell size and metabolism in response to nutrients, growth factors, and other extracellular cues [21]. To investigate the relationship between mTORC1 signaling and the role of vimentin in cell size, we analyzed mTORC1 activity after overnight serum starvation of cells and subsequent stimulation with insulin or FCS for 15 minutes. Intriguingly, we found that already without stimulation, the phosphorylation of mTORC1 downstream targets were lower in Vim −/− MEFs. Strikingly, Vim −/− MEFs stimulated with insulin or FCS showed only negligible mTORC1 activation (Fig 3D), indicating that the observed faulty cell size signaling is due to a defect in mTORC1 activation. Together, these results show that the cell size reduction in Vim −/− cells is due to abrogated mTORC1 signaling.

## Vimentin modulates the mTORC1 pathway through nutrient and insulin signaling

Along with insulin and growth factor signaling, mTORC1 is activated by nutrients [6]. To understand how the combination of nutrients and insulin signaling modulates vimentin-mediated mTORC1 signaling, we starved the cells for 1 hour in culture media without amino acids, glucose, and serum, followed by stimulation with amino acids L-glutamine with minimum essential amino acids (EAAs) alone or with non-EAAs, glucose, and insulin. As Vim −/− MEFs displayed a striking defect in the insulin-mediated stimulation of mTORC1 signaling, we first studied what are the differences in the presence of insulin while the nutrient sources are varied (Fig 4A). By maximal stimulation of insulin-mediated signaling, we wanted to reveal possible differential effects of variable nutrient sources and, in this way, examine the upstream parts of the insulin signaling pathway. The lack of a difference in AKT phosphorylation (Fig 4A), implies that vimentin does not regulate mTORC1 signaling in the upstream parts of the pathway. Importantly, when examining the downstream targets, we saw that when insulin signaling is pushed to the maximum, the downstream signaling is significantly suppressed regardless of the nutrient source (Fig 4A). This outcome demonstrates that the capacity of insulin signaling to amplify the mTORC1 pathway is always dependent on vimentin, regardless of which additional stimuli are provided by individual nutrients.

When examining the effect of nutrients alone, the overall finding is that in Vim −/− fibroblasts, there is very little signaling passing down to the targets downstream of mTORC1 (Fig 4B). While the addition of glucose to WT MEFs boosted the amino acid-mediated activation of P70S6K and the ribosomal protein S6, this amplification was basically absent in Vim −/− MEFs (Fig 4B). Although, the S6 activation stayed overall at very low levels in the Vim −/− MEFs, some residual S6 activation could be observed, with slight variation between different nutrient treatments (Fig 4B). The phosphorylation level of the mTORC1 downstream target 4EBP1 is also reduced in Vim −/− MEFs (Fig 4B). 4EBP1 is an inhibitory protein, whose functions are regulated by mTORC1-mediated phosphorylation. In a hypophosphorylated state, 4EBP1 blocks protein synthesis by binding to the eIF4E complex. mTORC1 inhibits this interaction through phosphorylation and promotes protein synthesis [2,22]. Importantly, we were able to rescue the phosphorylation of S6 in Vim −/− MEFs by transfecting WT vimentin (S1C–S1E Fig). Together, these results imply that vimentin deletion abrogates mTORC1 signaling impairing the phosphorylation of both S6K and 4EBP1, which are both likely to contribute to the reduced protein synthesis rates in the Vim −/− cells, with the consequent reduction in cell size.

To verify that only mTORC1 downstream signaling is affected, we performed the same experiment on cells treated with 100 nM of rapamycin, an inhibitor of mTORC1. We found that this treatment inhibits mTORC1 activation in WT MEFs and the residual mTORC1

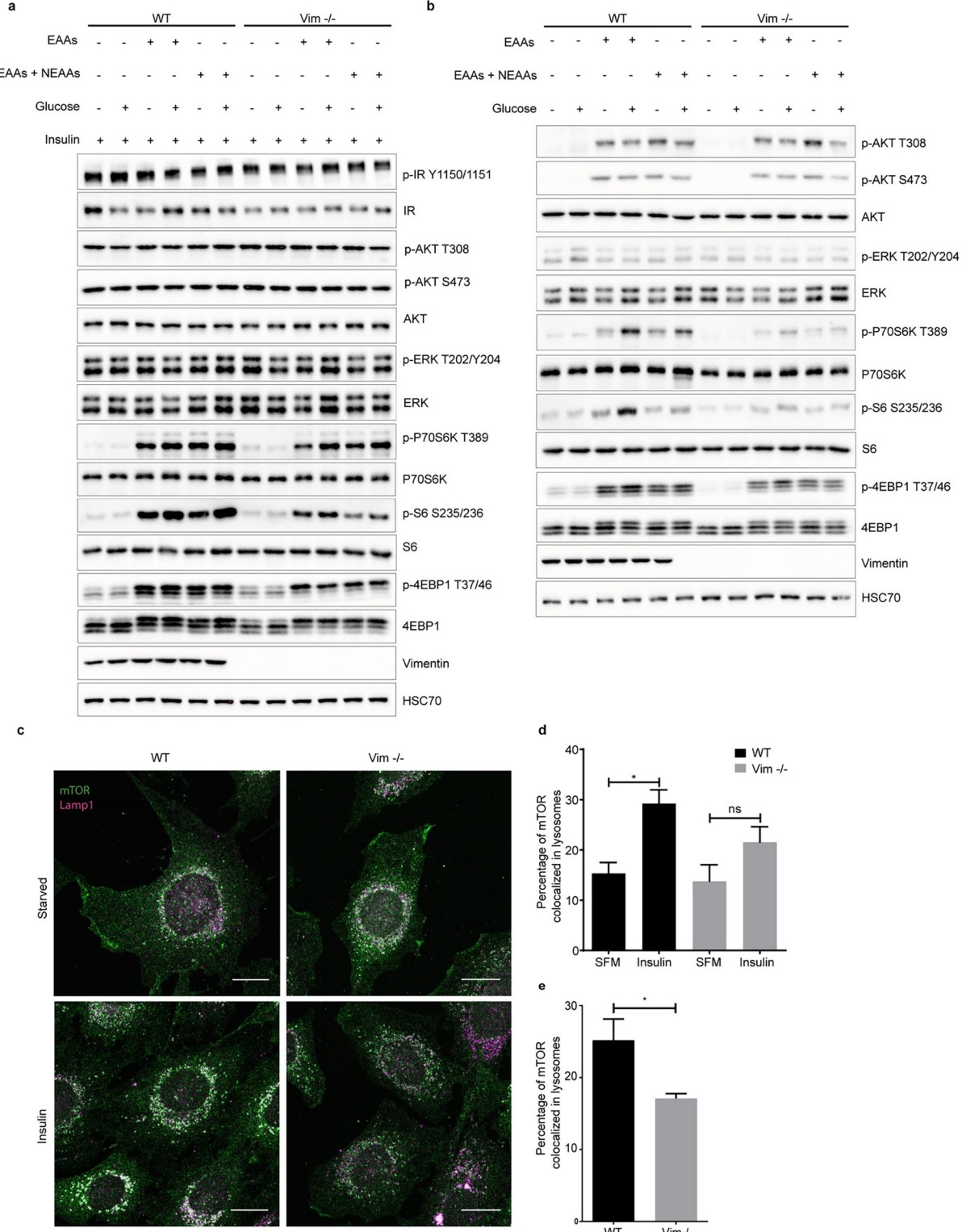

**Fig 4. Vimentin modulates the mTORC1 pathway through nutrient and insulin signaling.** (**a**) Western blot analysis of WT and Vim −/− MEFs starved with RPMI media lacking amino acids, glucose, and growth factors for 1 hour, followed by a 30-minute stimulation with 1 mM L-glutamine plus 1× EAAs alone or with NEAAs, plus 2.5 mM glucose and 100 nM insulin (*n* = 3). (**b**) Same experiment as in (**a**) performed without insulin. (**c**) Representative images of WT and Vim −/− MEFs serum starved overnight and stimulated with 100 nM insulin for 15 minutes. The cells were stained for mTOR and the lysosomal marker LAMP1 (scale bar = 20 μm). (**d**) The colocalization

analysis is represented as mean ± SEM of the percentage of colocalized pixels before and after insulin stimulation and it was performed using BioImageXD ($n = 3$). (**e**) Same as in (**d**) but in steady-state conditions. *$p < 0.05$, ns = nonsignificant. The data underlying the graphs shown in Fig 4D and 4E can be found in S1 Data. SFM, serum-free media; EAAs, essential amino acids; NEAAs, Non essential amino acids; mTORC1, mechanistic target of rapamycin kinase complex 1.

activation in Vim −/− MEFs (S2A and S2B Fig), validating that vimentin modulates mTORC1 activation by regulating its capability to activate downstream effectors. Altogether, these results show that vimentin is required for the phosphorylation of mTORC1 downstream effectors to take place, pointing to a role of vimentin in mTORC1 activation itself.

S6 is phosphorylated at serine 234/235 through P70S6K, which is downstream of the AKT-mTORC1 signal transduction pathway [21]. S6 can also be activated by P90S6K, a downstream effector of ERK signaling [23]. Therefore, we used AKT and ERK kinase inhibitors to determine whether these kinases play a role in amplifying S6 phosphorylation in a vimentin-dependent manner. We treated WT and Vim −/− MEFs with either an ERK or AKT inhibitor after 40 minutes of full starvation and performed the nutrient and insulin stimulation protocol described before. In both WT and Vim −/− MEFs, AKT inhibition completely blocked AKT and its downstream mTORC1 signaling, including phosphorylation of P70S6K, and S6 (S2C Fig). However, ERK inhibition had no effect on mTORC1 downstream signaling (S2C Fig). Thus, vimentin modulates mTORC1 downstream signaling independently of ERK signaling and the phosphorylation of S6 requires an active mTORC1.

## mTOR translocation to lysosomes is impaired in Vim −/− MEFs

Since the effect of vimentin was narrowed down to direct targeting of mTORC1, we wanted to examine the mechanisms underlying this effect. In this respect, mTORC1 translocate from the cytosol to lysosomes for its activation [5,24]. This is an essential step for mTORC1 activation by Rags [24]. Therefore, we measured mTORC1 translocation in serum-starved MEFs upon insulin stimulation using confocal imaging and colocalization analysis. While insulin stimulation strongly increases colocalization of mTOR with the lysosomal marker LAMP1 in WT MEFs, there was no significant colocalization increase in Vim −/− MEFs (Figs 4C, 4D and S3A). Moreover, also at steady-state conditions, colocalization of mTOR to lysosomes is significantly reduced in Vim −/− MEFs, when compared to their WT counterpart (Figs 4E and S3B). This demonstrates that the translocation machinery requires vimentin. Thus, vimentin has the ability to support mTORC1 localization to the lysosomal membrane. This modus operandi is in agreement with the established signaling functions of vimentin, which has been identified as a scaffolding system for many signaling events and pathways [25–28].

## Constitutively active Rag GTPase rescues mTOR localization in Vim −/− MEFs

The level of amino acids in the cytosol and lysosomes determine mTORC1 translocation and activation. Rag GTPases sense the levels of amino acids in the cytoplasm and lysosomal lumen [5,29], leading to recruitment of mTORC1 to lysosomes for activation [29]. To understand the molecular mechanisms underlying the vimentin-mediated mTORC1 translocation, we measured the colocalization of mTOR on the lysosomal membrane upon transfection of WT Rag GTPase or constitutively active Rag GTPase, the latter which recruits mTOR to lysosomes even in the absence of amino acids [5,30]. Vim −/− MEFs shows significantly lower mTOR colocalization to the lysosome membrane than WT upon expression of exogenous WT Rag GTPase (Figs 5A, 5B and S4). Importantly, expression of constitutively active Rag GTPase rescued the mTOR colocalization in Vim −/− MEFs (Figs 5A, 5B and S4). Based on these results,

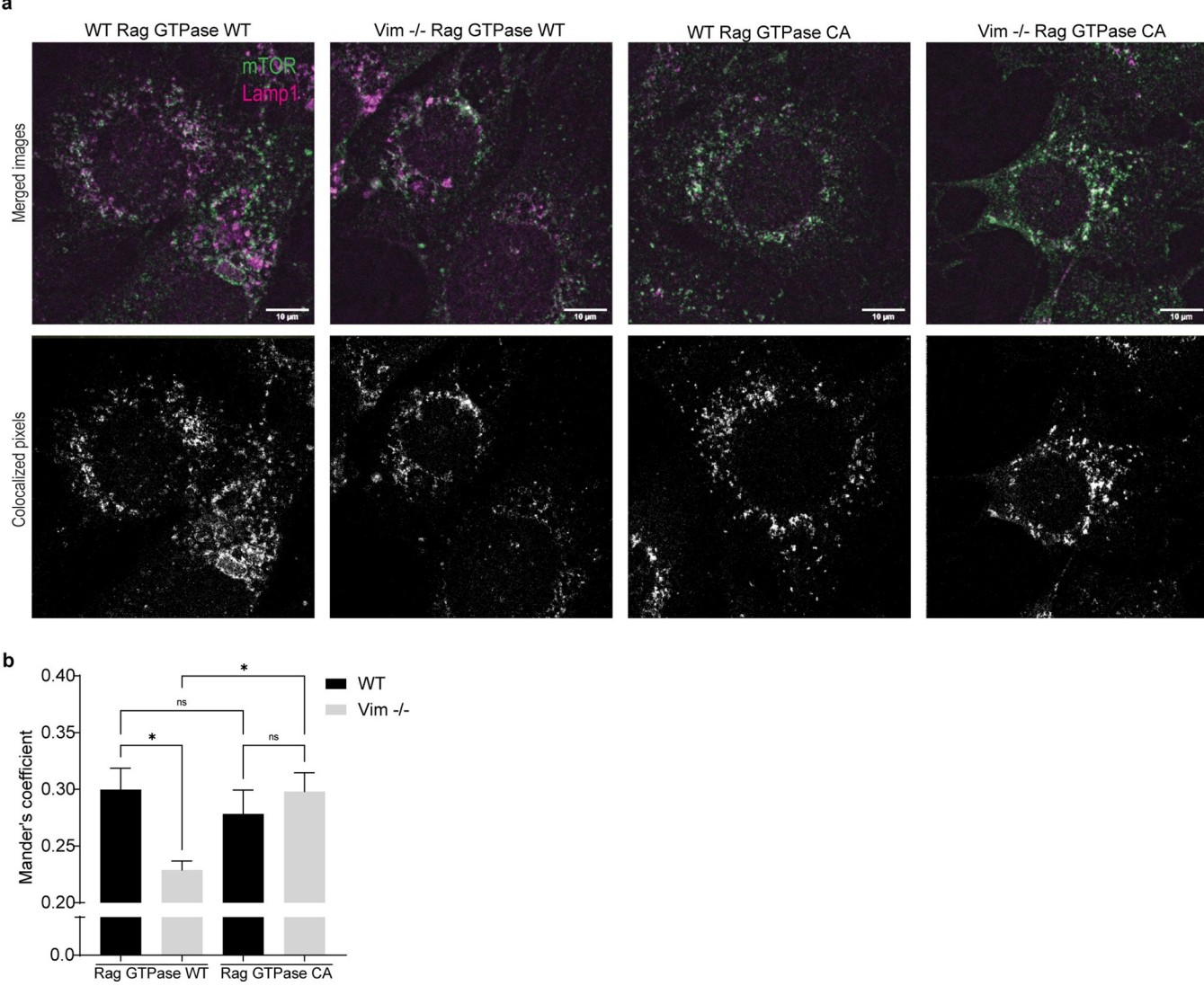

**Fig 5. Constitutively active Rag GTPase rescues mTOR localization in Vim −/− MEFs.** (**a**) mTOR and LAMP1 were stained in WT and Vim −/− MEFs transfected with either normal Rag GTPase or constitutively active Rag GTPase. Representative images show the merged channels of mTOR and LAMP1 (scale bar = 10 μm). (**b**) Quantification of mTOR and LAMP1 colocalization measured from the images in (a) using Fiji. Mander's coefficient is represented as mean ± SEM of mTOR colocalization on lysosome membrane ($n = 3$). $^*p < 0.05$, ns = nonsignificant. The data underlying the graphs shown in the Fig 5B can be found in S1 Data.

vimentin determines mTORC1 signaling by regulating Rag GTPase activation. Thus, vimentin could affect nutrient sensing, through which it modulates mTORC1 signaling.

## Vimentin inhibits autophagy through mTORC1

As vimentin has been shown to regulate autophagy [15] and autophagy is also closely coupled to mTORC1 signaling, we wanted to determine whether the results we obtained with mTORC1 signaling could be coupled to the vimentin-mediated regulation of autophagy. During autophagy, LC3I, a cytosolic protein, is conjugated with phosphatidylethanolamine to form LC3-II, which is recruited to phagophores, then moved into autophagosomes and is finally degraded in autolysosomes. Thus, the autophagic flux can be obtained by measuring

the rate of conversion of LC3I to LC3II [31]. p62 interacts with autophagy substrates and it selectively transports ubiquitinated proteins to lysosomal cargo for autophagy, a process during which p62 also gets degraded. Therefore, inhibition of autophagy will increase p62 levels, while activation of autophagic flux will reduce the p62 levels [32]. Importantly, we observed that nutrient starvation significantly decreased p62 levels in Vim −/− MEFs, suggesting activation of the autophagic flux, with prominently lower p62 levels as compared to the samples from starved WT MEFs (Fig 6A). In Vim −/− MEFs, this could be partly inhibited in cells stimulated with insulin, amino acids, and glucose (Fig 6A). To further investigate the role of vimentin in autophagy, we subjected cells to depletion of various nutrient sources (glucose, EAAs or EAAs and NEAAs). WT MEFs appeared to be well protected against autophagic flux, as reflected by the steady levels of p62. Strikingly, in Vim −/− MEFs, the levels of p62 are significantly lower than in all the corresponding WT MEF samples, indicating that without vimentin, the cells are significantly more susceptible to autophagy (Fig 6B). Notably, the p62 depletion data can be coupled to the inhibition of the mTORC1 pathway (Fig 6B). To further strengthen these observations and to determine autophagic flux, we measured the conversion of the LC3I and LC3II by using a pMRX-IP-GFP-LC3-RFP-LC3ΔG vector. GFP-LC3 undergoes cleavage and enters lysosome during autophagy; however, RFP-LC3ΔG lacks the cleavage site and remains in the cytosol. In this way, the GFP/RFP fluorescent intensity ratio is a measure of the autophagic flux [33]. While starvation, surprisingly, increases the conversion of LC3I to LC3II in both WT and Vim −/− MEFs (Figs 6C, S5A and S5B), in Vim −/− MEFs, the rate of conversion is significantly faster. This suggests that Vim −/− MEFs cannot withstand limited nutrient levels and are more prone to autophagy than WT MEFs. Importantly, our results show that WT MEFs can sustain p62 levels even in the absence of insulin and that, conversely, Vim −/− MEFs activate autophagy even in the presence of nutrients (Fig 6B). Thus, the inhibition of autophagy we observe here takes place by mechanism different from the AKT and 14-3-3-dependent mechanism previously described [15], in which vimentin and Beclin 1 are phosphorylated by AKT [15]. This notion is especially relevant as in these experiments following nutrient starvation, we saw no difference in AKT activation [15]. It is well established that mTORC1 can be activated by nutrients alone [6] and that mTORC1 inhibits autophagy by phosphorylating ULK1 [8], the autophagy-activating kinase that is required to trigger autophagy by Beclin-1 phosphorylation [9]. To determine to what extent vimentin also regulates cell size through mTORC1-mediated regulation of autophagy, we assessed the ULK1 serine 757 phosphorylation level, the phosphorylation of which is mediated by mTORC1 to inhibit autophagy. We found that this phosphorylation level is reduced in Vim −/− MEFs, whereas in WT MEFs the phosphorylation is increased and sustained, both under stimulation (Fig 6A) and nutrient limitation (Fig 6D). This implies that vimentin controls autophagy through mTORC1-mediated phosphorylation of ULK1, leading to its inhibition. To examine this pathway in greater detail, we also tested whether normal cell volume could be rescued in Vim −/− MEFs using dominant-negative ULK1 K46N, which inhibits autophagy independently of mTORC1 [34]. Overexpressing dominant-negative ULK1 K46N did not rescue cell size in Vim −/− MEFs (S6 Fig). These results demonstrate that, although the Vim −/− are more prone to autophagy due to reduced mTORC1 activity, inhibiting autophagy through dn-ULK1 is not sufficient to increase cell size, but it would require up-regulation of mTORC1 activity. These results show that the vimentin-mediated regulation of mTORC1 activation will also control autophagy, but that autophagy alone is not the primary reason for the reduced cell size in Vim −/− cells but rather the inhibited protein synthesis due to inhibition mTORC1 signaling. Therefore, vimentin regulates both mTORC1-mediated protein synthesis as well as autophagic flux by mTORC1-mediated ULK1 inhibition.

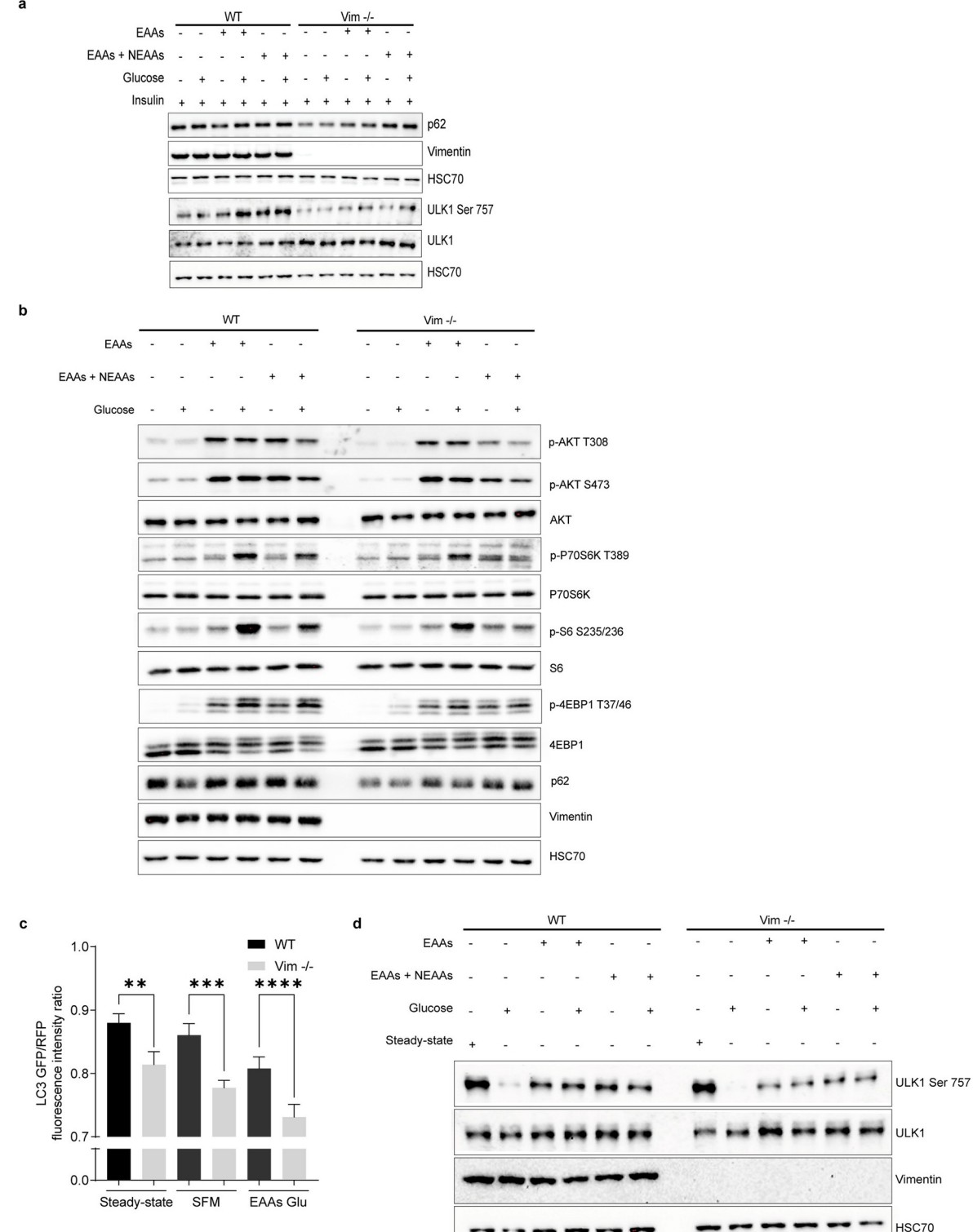

**Fig 6. Vimentin protects MEFs from autophagy.** (**a**) Western blot analysis of WT and Vim −/− MEFs starved with RPMI media lacking amino acids, glucose, and growth factors for 1 hour, followed by a 30-minute stimulation with 1 mM L-glutamine plus EAAs alone or with NEAAs and 2.5 mM glucose, 100 nM insulin (*n* = 3). (**b**) Western blot analysis of WT and Vim −/− MEFs grown 1 hour in RPMI media with different nutrients (1 mM of L-glutamine plus EAAs alone or with NEAAs and 2.5 mM glucose; *n* = 3). (**c**) Autophagic flux measured through LC3 GFP/RFP fluorescent intensity ratio in cells transfected with 7.5 μg of pMRX-IP-GFP-LC3-RFP-LC3ΔG. Graph represents

mean ± SEM of LC3 GFP and RFP fluorescence intensity ratio of WT and Vim −/− MEFs under normal conditions (steady state, DMEM with serum) or with either SFM or RPMI media lacking amino acids, glucose, and growth factors supplemented with 2 mM of L-glutamine, 1× EAAs and 2.5 mM glucose for 3 hours. Fluorescent intensities were measured with Fiji ($n = 3$). (**d**) Western blot analysis of WT and Vim −/− MEFs grown for 1 hour in normal DMEM with serum or RPMI media lacking amino acids, glucose, and growth factors supplemented with nutrients in different combinations (1 mM L-glutamine plus 1× EAAs alone or with NEAAs and 2.5 mM glucose, $n = 3$). $^*p < 0.05$, ns = nonsignificant. The data underlying the graphs shown in Fig 6C can be found in the S1 Data. EAAs, essential amino acids; NEAAs, Non essential amino acids; SFM, serum-free media.

## Conclusions

Here, we unravel a novel role of vimentin in cell size regulation, as the whole cell size machinery seems to be guarded and concerted by the presence of vimentin. Moreover, we provide evidence that vimentin regulates these processes by mediating mTORC1 lysosomal localization and activation through Rag (Fig 7). We also show that vimentin thereby also regulates autophagy by mTORC1-mediated inhibition of the autophagy-activating kinase ULK1. During wound healing and regeneration, the presence of nutrients and growth factors is crucial for cell growth and proliferation. Fibroblasts are rapidly proliferating cells that are crucial for the formation of granulated tissue in injury areas [35]. In these dynamic but stressful conditions, vimentin levels and its organization are actively coordinated. Our results show that vimentin is the critical link between cell size signaling and other processes critical for repair. The advantage of a vimentin-based link to cell size signaling is that vimentin is also coupled to cytoskeleton-mediated sensing of cell shape [36] and to interactions with both the surface and other cells [11]. High expression of vimentin is a hallmark of many types of cancer cells. Providing an active regulation of cell size coupled to resistance to autophagy is, obviously, an additionally important and advantageous function of vimentin when present in cells in dynamic transition or in cancer cells.

## Materials and methods

### Plasmids and antibodies

The pCMV-script-vimentin plasmid was a kind gift from Professor Johanna Ivaska [37]. pCMV4-HA was a gift from Shao-Cong Sun (Addgene plasmid # 27553; http://n2t.net/addgene:27553; RRID:Addgene_27553) [38]. pME18S-3HA-mULK1(K46N) was a gift from Masaaki Muramatsu (Addgene plasmid # 22897; http://n2t.net/addgene:22897; RRID: Addgene_22897) [39]. The antibodies used for the experiments are listed in S1 Data.

### Mice and ethical statement

WT and Vim −/− mice were generated from heterozygous mice by crossing 129/SV × C57BL/6 strains [14]. The mice were held at the central animal laboratory, Biocity unit (KEK/2010-2112-Eriksson) and it follows good laboratory practice and it routinely inspect by Finish Medicines Agency (Fimea) to monitor standard operating procedures. Mice were fed with standard diet and free access to water. Mice were sacrificed by cervical dislocation. All experiments were performed according to the guidelines set by the Ethical Committee of central animal laboratory. Genotyping was determined by polymerase chain reaction. The body weight was measured at 3 ($n = 6$) and 6 ($n = 4$) months old. BMI and Lee's index of 6 months old mice ($n = 8$) were calculated accordingly. The total fat mass was measured by EcoMRI and normalized to the body weight ($n = 7$ to 8). To evaluate organ size and weight, WT and Vim −/− mice of 3 months of age ($n = 3$) were sacrificed. The white adipose tissues were fixed in 4% paraformaldehyde (PFA) and embedded in paraffin. The sectioning and hematoxylin-eosin staining was carried out by Lounais-Suomen pathology laboratory. The sections were imaged using the

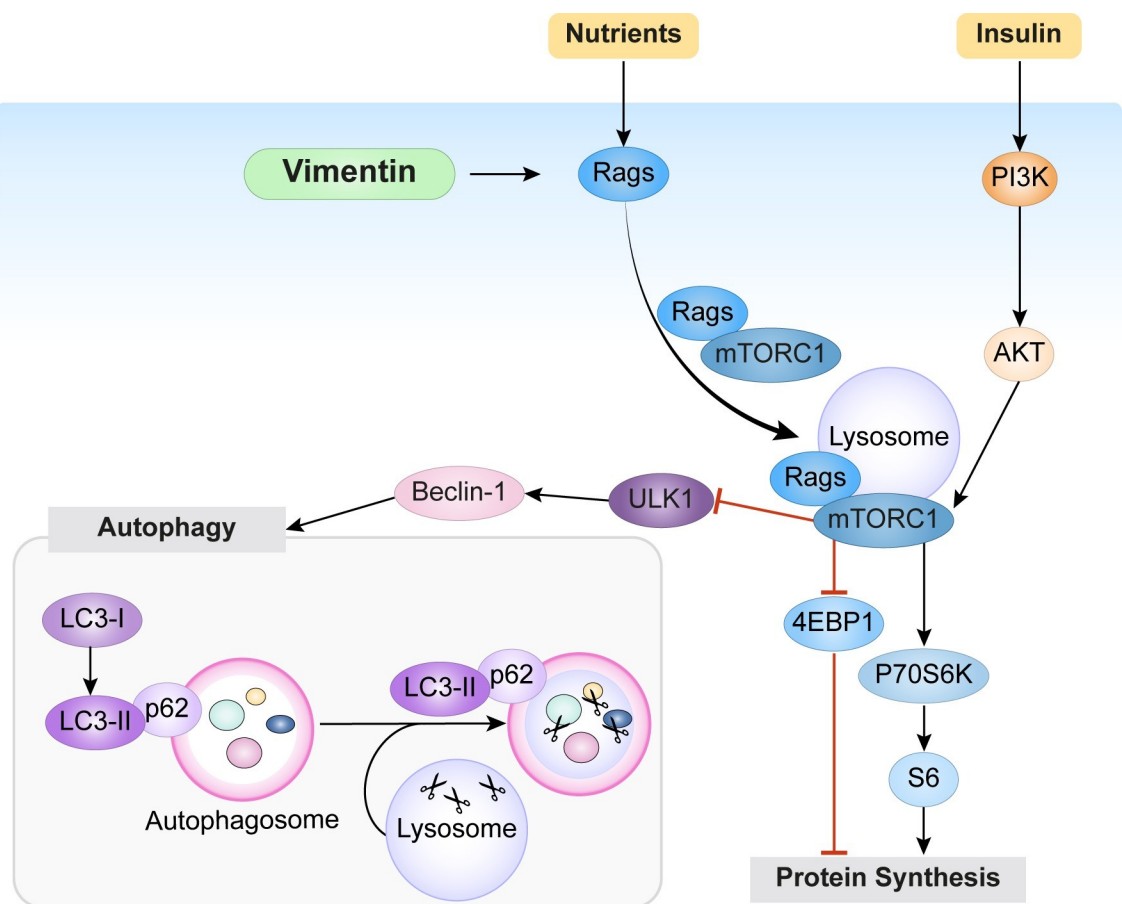

**Fig 7. Vimentin regulates mTORC1 signaling by facilitating mTORC1 localization to lysosomes.** A simplified scheme of the mTORC1 signaling system. Our results demonstrate that in the presence of nutrients, vimentin facilitates mTORC1 activation by regulating Rag GTPase activation. This results in the phosphorylation of 4EBP and P70S6K downstream targets, which lead to protein synthesis. Moreover, activation of mTORC1 results in decreased autophagy, as demonstrated by higher levels of the autophagic flux indicator p62 and reduced LC3I conversion to LC3II. As this autophagy inhibition works even under nutrient-starved condition, in the absence of AKT activation, the results imply that vimentin inhibits autophagy by mTORC1-mediated inhibition of ULK1-Beclin pathway. In this way, vimentin promotes cell size in 2 directions: by boosting protein synthesis through mTORC1 activation and by inhibiting of protein degradation through inhibition of autophagic flux. mTORC1, mechanistic target of rapamycin kinase complex 1.

Panoramic Slide Scanner (3DHISTECH, Hungary) and Fiji was used to measure cell size ($n = 3$, 250 cells in total).

## Cell culture and treatments

SV40-immortalised WT and Vim −/− MEFs were cultured in Dulbecco's modified media (DMEM, Sigma #D6171) supplemented with 10% fetal bovine serum (FBS, Biowest #S1810), 2 mM of L-glutamine (Biowest #X0550), 100 U/ml of penicillin, and 100 μg/ml of streptomycin (Sigma #P0781) at 37˚C in a 5% $CO_2$ incubator and passaged when they were about 80% confluent.

To test if the size phenotype could be rescued, Vim −/− MEFs were seeded in 6-well plates on the day prior to transfection. Using XfectTM (Clontech Laboratories, United States of America #631318) 7.5 μg of pCMV script-vimentin, pCMV4-HA and pME18S-3HA-mULK1 K46N were transfected separately in Vim −/− MEFs according to instructions of the manufacturer. The cells were used for cell volume analysis 48 hours post-transfection.

To assess how growth factors affect the cell size in WT and Vim −/− MEFs, the cells were incubated in serum-free media (SFM) for 2 to 3 days. Then, the media was replaced with fresh SFM, and the cells were stimulated either 100 μg/ml insulin (Sigma #I0516) or 5% fetal calf serum (FCS, Biowest #S1710) for 24 hours. To study mTORC1 activation, 200,000 cells were seeded in 6-well plates and serum starved overnight. Next, the cells were treated either with 100 nM insulin or 5% FCS for 15 minutes. Alternatively, the cells were fully starved for 1 hour with RPMI media (Life Science #R9010-01) lacking glucose, amino acids, and serum. Then, the cells were treated for 30 minutes with 1 mM of L-glutamine plus 1× concentration of either minimum EAAs, (Thermo Fisher Scientific #11130036) or EAAs and non-essential amino acids (NEAAs) mix (1:1 mix of EAAs with NEAAs, Sigma #M7145) either in the presence or absence of 2.5 mM of glucose (Thermo Fisher Scientific #A2494001) for 30 minutes. This experiment was repeated with 100 nM of insulin along with amino acids and glucose. To further investigate mTORC1 signaling, the cells were treated with 100 nM of rapamycin (Tocris #1292) for 20 minutes. This was conducted after 40 minutes of full starvation and was followed by stimulation with amino acids, insulin, and glucose as described above. Alternatively, 2 other kinase inhibitors were used: 1 μm and 1.5 μm of AKT VIII (Calbiochem #124018) or 100 nM and 200 nM of trametinib (Selleckchem #GSK1120212). To study autophagy, 200,000 WT and Vim −/− were seeded in 6-well plates and incubated for 16 hours. On the next day, the media was replaced with RPMI media containing 1 mM glutamine plus EAAs, with or without NEAAs and glucose for 1 hour. The protein levels were evaluated using immunoblotting. To study the role of Rag GTPase in mTOR translocation, WT and Vim −/− MEFs were transfected with either 4.5 μg of pRK5-HA GST RagB WT (Addgene plasmid #19301; http://n2t.net/addgene:19301; RRID:Addgene_19301) and pRK5-HA GST RagC WT (Addgene plasmid #19304; http://n2t.net/addgene:19304; RRID:Addgene_19304) or 4.5 μg of pRK5-HA GST RagB 99L (Addgene plasmid #19303; http://n2t.net/addgene:19303; RRID:Addgene_19303) and pRK5-HA GST RagC 75L (Addgene plasmid #19305; http://n2t.net/addgene:19305; RRID:Addgene_19305) using Xfect (Clontech Laboratories, USA) according to manufacturer's instructions [5].

## Primary MEFs isolation

The primary MEFs were isolated according to a previously described method [40]. Briefly, the mice were sacrificed 13 to 14 days post-coitum by cervical dislocation, the uterine horns were dissected, rinsed with 70% ethanol, and kept in PBS. Under the laminar hood, the embryos were separated, and both head and red organs of the embryos were stored for genotyping. The remaining tissue was washed with PBS and minced until it became possible to pipette it. The tissues were incubated for 15 minutes at 37˚C with 2 ml of 0.05% trypsin/EDTA containing 100 K units of DNaseI, and the cells were dissociated by pipetting every 5 minutes. Trypsin was inactivated by adding about 1 volume of freshly prepared DMEM complete media and the cells were centrifuged (100 g) for 5 minutes. The supernatant was carefully removed; the cell pellet was resuspended in pre-warmed DMEM complete media and plated approximately 1 embryo per plate.

## Bone marrow-derived dendritic cells (BMDCs)

Primary dendritic cells were differentiated from bone marrow precursors isolated from WT and Vim −/− mice. The differentiation was performed using IMDM medium containing FCS (10%), glutamine (20 mM), penicillin–streptomycin (100 U/ml), 2-ME (50 μm) further supplemented with granulocyte-macrophage colony-stimulating factor (50 ng/ml)-containing supernatant obtained from transfected J558 cells, as previously described [41]. The semi-adherent

cell fraction, corresponding to the CD86$^+$ dendritic cells, was retrieved from the cell culture dishes at the differentiation days 10 to 12 by gentle flushing and used for the measurements. All cell culture reagents were purchased from Thermo Fisher Scientific (Waltham, Massachusetts, USA).

## Cell area and diameter

Phase contrast images from WT and Vim −/− MEFs were taken using CellIQ (ChipMan Technologies, Finland) using a 10× objective. The cell area was manually measured using the freehand tool in Fiji ($n$ = 3, 200 cells in total) [42]. Cell area of WT and Vim −/− BMDCs was evaluated based on images taken for undeformed (inlet) or deformed (using 2 different flow rates; fr1 = 0.16 μl/s and fr2 = 0.32 μl/s) cells in a real-time deformability cytometry (RT-DC) setup [20] using 40× objective (EC-Plan-Neofluar, 40×/0.75; #420360–9900, Zeiss, Germany). The RT-DC measurements were performed according to previously established protocol [43] using a 30-μm channel. The cell contours were identified using thresholded images, and the cell cross-sectional area was derived from a convex hull of the fitted contours. After the treatment, WT and Vim −/− MEFs were trypsinized and imaged under Lecia DMIL light microscope using 10× objective. The cell diameter was manually measured using the freehand tool in Fiji ($n$ = 3, 50 cells in total).

## Cell volume

Immortalized and primary WT and Vim −/− MEFs were grown in 6-well plates overnight. The samples were harvested and fixed with 3% PFA for 15 minutes and then stained with 2 μm of CellTracker fluorescent probes (Thermo Fisher Scientific #C34552). Z-stacks were taken using a Leica TCS SP5 Matrix confocal microscope with the 20× objective. The cell volume was measured using Imaris software 8.1 (BITPLANE, Switzerland). Three clones of MEFs were used for the analysis ($n$ = 3, 1,800 to 1,900 cells in total). Cell volume analysis was also performed in cells treated for 24 hours with 1 mM thymidine ($n$ = 3, 250 to 400 cells) (Sigma #T1895) and in Vim −/− MEFs transfected with pCMV-script-vimentin ($n$ = 3, 1,000 to 1,300 cells). To measure cell volume in cells transfected with HA-tag-dnULK1 K46N or HA-tag without insert, cells were immunolabeled for the HA tag with a fluorescent dye Alexa Fluor 488 and stained with CellTracker Red CMTPX Dye (Thermo Fisher Scientific #C34552). Then, cells were imaged as we mentioned before and cell volume of transfected cells were measured by Fiji ImageJ using 3D object counter ($n$ = 3, 300 to 600 cells in total). For WT and Vim −/− BMDCs, the volume was calculated by rotation of the 2D cell contours obtained from RT-DC measurements. RT-DC data were reanalyzed from recently published measurements [44]. The calculation was performed using ShapeOut software (ShapeOut 0.9.5; https://github.com/ZELLMECHANIK-DRESDEN/ShapeOut; Zellmechanik Dresden, Germany) according to a previously described approach [45].

## Protein concentration

Three different WT and Vim −/− MEFs clones were seeded in 6-well plates and incubated overnight. Then, the cells were trypsinized and $3.0 \times 10^5$ cells were lysed for 1 hour at 4˚C in a buffer containing 150 mM NaCl, 1% Triton X-100, 0.2% SDS, 50 mM Tris (pH 8.0), and 0.5% of sodium deoxycholate. The protein concentration was measured using a BCA (Thermo Fisher Scientific #23227) according to the manufacturer's instructions. The protein concentration was normalized to the number of cells.

## Immunoblotting

WT and Vim −/− MEFs (treated as described above) were washed 3 times with ice cold PBS and lysed with 3× sample buffer (0.625 M Tris-HCL (pH 6.8), 3% sodium dodecyl sulfate, 30% glycerol, 0.015% bromophenolblue, 3% β-mercaptoethanol). The lysates were heated for 10 minutes at 98˚C. The samples were separated by SDS-PAGE, transferred to nitrocellulose membranes (or methanol-activated polyvinylidene difluoride membranes for LC3), and blocked 1 hour with 5% milk in TBS 0.3% Tween20. The membranes were incubated with primary antibody overnight at 4˚C. All primary antibodies used for western blot were diluted 1:1,000 in TBS 0.3% Tween20 3% BSA 0.02% sodium azide. The membranes were washed 20 minutes with TBS 0.3% Tween20 and incubated for 1 hour at room temperature with secondary antibody (diluted 1:10,000 in TBS 0.3% Tween20 5% milk). The antibody signals were probed with enhanced chemiluminescence (Amersham #RPN2236) and detected in a BioRad Chemidoc machine.

## Immunostaining

For the mTOR translocation assay, WT and Vim −/− MEFs were seeded on top of glass coverslips and serum starved overnight. The cells were treated with 100 nM of insulin for 15 minutes, were washed once with PBS, and were fixed with 3% PFA in PBS for 15 minutes at room temperature. Then, the cells were permeabilized with PBS 0.2% Triton X-100 for 5 minutes and blocked with 3% BSA in PBS 0.2% Tween20 for 1 hour. The coverslips were incubated overnight in a humidified chamber with primary antibody diluted 1:200 (mTOR), 1:20 (LAMP1), and 1:2,000 (vimentin) in PBS 3% BSA 0.2% Tween20. On the next day, the coverslips were washed 3 times with PBS 0.2% Tween20, for 5 minutes each, and incubated with secondary antibody in PBS 3% BSA 0.2% Tween20 for 1 hour in the dark, at room temperature. The coverslips were washed as described before and dipped in Milli-Q water once. The excess water was removed, the coverslips were mounted on the slides with mowiol 4–88 and left to dry overnight. The imaging was performed using a Leica SP5 TCS confocal microscope with the 63× immersion oil objective numerical aperture 1.32. The colocalization analysis of LAMP1 and mTOR ($n$ = 3, 12 to 15 cells per trial) was carried out with the colocalization tool on BioImageXD [46]. For the Rag activity experiment, cells were seeded on glass cover slips 24 hours after transfection and, on the next day, cells were fixed with 3% PFA. Cells were stained and imaged as described above. The antibody dilutions were as follows: 1:200 (mTOR), 1:20 (LAMP1), and 1:200 (HA-tag). Fiji JaCoP tool was used for the colocalization analysis of mTOR and LAMP1. This was performed in cells which expressed exogenous Rag GTPase by calculating the Mander's coefficient ($n$ = 3, 8 to 12 cells per repeat).

## Protein synthesis assay

WT and Vim −/− MEFs were seeded in 96-well plates and serum starved overnight. The assay was carried out using the Click-iT HPG Alexa Fluor Protein Synthesis Assay Kits (Thermo Fisher Scientific #C10428). Briefly, the media was changed to methionine-free media with 50 μm of a methionine analogue. The cells were treated with 100 nM of insulin for 30 minutes and were washed once with PBS. The cells were fixed with 3% PFA in PBS and permeabilized with PBS 0.3% Triton X-100. The Click-it reaction was prepared according to the instructions of the manufacturer and added to the cells for 30 minutes in dark. Then, the cells were washed and kept in PBS. The imaging was performed with Cell-IQ (Chip-Man Technologies, Finland) using the green channel and the 10× objective. The image analysis ($n$ = 3, 100 cells in total) was carried out with the Cell-IQ Analyzer software by measuring the signal intensity.

## Autophagy flux

To measure the autophagic flux, WT and Vim −/− MEFs were seeded in 6-well plates and allowed to grow overnight. Then, 7.5 μg of the pMRX-IP-GFP-LC3-RFP-LC3ΔG vector (Addgene plasmid # 84572; http://n2t.net/addgene:84572; RRID: Addgene_84572) was transfected using Xfect (Clontech Laboratories, USA) according to manufacturer's instructions. After 24 hours, cells were seeded on glass cover slips and, on the next day, switched to SFM (DMEM, Sigma #D6171) or RPMI media (Life Science #R9010-01) supplemented with 1 mM of L-glutamine plus minimum EAAs mix (Thermo Fisher Scientific #11130036) and 2.5 mM glucose (Thermo Fisher Scientific #A2494001) for 3 hours. Cells were fixed with 3% PFA for 15 minutes and imaged with Leica SP5 TCS confocal microscope under a 63× immersion oil objective with a numerical aperture of 1.32. In transfected cells, fluorescent intensity of GFP and RFP were measured with Fiji using the free hand tool, and ratios of GFP/RFP fluorescent intensity were calculated as a measure of autophagic flux ($n$ = 3, 100 to 120 cells in total).

## Flow cytometry

Vim −/− MEFs were transfected as described in previously to assess whether mTORC1 activity could be rescued. After 24 hours, cells were fully starved for 1 hour with RPMI media without amino acids, glucose, and serum (Life Science #R9010-01). Then, cells were treated for 30 minutes with 1 mM of L-glutamine plus 1× concentration of minimum EAAs mix (Thermo Fisher Scientific #11130036), 2.5 mM of glucose (Thermo Fisher Scientific #A2494001), and 100 nM insulin. Then, cells were fixed with 3% PFA for 20 minutes and permeabilized with 0.2% Triton X-100 for 10 minutes. Cells were incubated in a primary antibody against S6 Ser235/236 (1:100) in PBS 3% BSA 0.2% Tween20 for 1 hour and washed twice with PBS 0.2% Tween20. Samples were incubated with secondary antibody (anti-rabbit Alexa 488, 1:500) in PBS 3% BSA 0.2% Tween20 for 30 minutes and then washed twice. Cells were labeled with vimentin Alexa Fluor 647 for 30 minutes in PBS 3% BSA 0.2% Tween20 and washed twice. Data acquired with BD LSRFortessa (BD Biosciences) using FITC 530/30 (515 to 545 nm) and APC 670/14 (663 to 677 nm). Data were analyzed with FlowJo by measuring the median fluorescent intensity of S6 Ser235/236 in vimentin positive and vimentin null cells gated populations.

## Statistical analysis

All the statistical analysis was carried out with GraphPad Prism 7 (GraphPad Software, USA). Three independent trials were carried out for each experiment unless it is stated otherwise. The statistical significance between 2 groups were measured by unpaired Student $t$ test with Welch correction. The Mann–Whitney test was used for samples not following a normal distribution. Multiple comparisons were performed using either 1-way ANOVA or Welch ANOVA test with Holm–Sidak correction for multiple testing. For BMDC measurements, 5 independent experiment replicates were performed, and the statistical analysis was performed using linear mixed effect model in ShapeOut according to previously described procedures [45].

## Supporting information

**S1 Video Video on the effect of vimentin on cell size Live cell imaging shows that Vim −/− MEFs are consistently smaller than WT MEFs grown to confluency.**
(MP4)

**S1 Fig. The size phenotype is not restricted to MEFs. (a**) Cell area of WT (black) and Vim −/− (gray) BMDCs obtained from RT-DC measurements of undeformed, spherical cells

(initial) and cells deformed in a narrow constriction of a microfluidic channel using 2 different flow rates (fr1 = 0.16 μl/s and fr2 = 0.32 μl/s). (**b**) Cell volume of WT (black) and Vim −/− (gray) BMDCs corresponding to the samples in (**a**). In (**a**) and (**b**), each data point represents a mean of an independent RT-DC measurement ($n$ = 5), with at least 1,000 cells evaluated per measurement. $^*p < 0.05$, $^{**}p < 0.01$. RT-DC cell size data has been obtained by reanalyzing recently published measurements (44). (**c**) Dot plot of phospho-S6 fluorescent intensity in WT (MFI = 28,274) and Vim −/− MEFs (MFI = 9,906). (**d**) Corresponding dot plot analysis of phospho-S6 fluorescent intensities of Vim −/− MEFs starved (MFI = 895) and nutrient stimulated (MFI = 3,214) compared with (**e**). Vim −/− MEFs transfected with WT vimentin starved (MFI = 29,588) and nutrient stimulated (MFI = 39,099). Nutrient stimulation was done by starving cells for 1 hour without nutrient and growth factors followed by stimulation with 1× EAAs, 1 mM glutamine, 2.5 mM glucose, and 100 nM insulin for 30 minutes. MFI = median fluorescent intensity of Alexa Fluor 488. One representative of several. Gating used for the analysis can be found in the Figshare https://doi.org/10.6084/m9.figshare.20024534.v1. The data underlying the graphs shown in the S1 Fig can be found in S1 Data.
(TIF)

**S2 Fig. Phosphorylation of mTORC1 downstream targets requires an active mTORC1.** (**a**) Western blot analysis of WT and Vim −/− MEFs starved with RPMI media lacking amino acids, glucose, and growth factors for 1 hour, followed by stimulation with EAAs or EAAs and NEAAs, with or without 2.5 mM glucose in all combinations. Cells were treated with 100 nM of rapamycin after 40 minutes of starvation ($n$ = 3). (**b**) Same experiment but including a 100 nM insulin treatment ($n$ = 3). (**c**) Western blot analysis of WT and Vim −/− MEFs treated as in S2B Fig in the presence of an ERK (TRAM) or an AKT (AKT VIII) inhibitor ($n$ = 3).
(TIF)

**S3 Fig. mTOR colocalization to lysosomes is higher in WT MEFs.** (**a**) WT and Vim −/− MEFs were serum starved overnight and stimulated with 100 nM insulin for 15 minutes. Cells were stained with mTOR (green channel) and the lysosomal marker LAMP1 (far red channel). (**b**) Same as in (a), but in steady state conditions (scale bar = 20 μm).
(TIF)

**S4 Fig. Constitutively active Rag GTPase rescues mTOR localization to lysosomes in Vim −/− MEFs.** Images show mTOR, HA tagged WT Rag GTPase, or HA tagged constitutively active Rag GTPase and LAMP1 in WT and Vim −/− MEFs. Merged images represent mTOR and LAMP1 channels.
(TIF)

**S5 Fig. Autophagic flux is higher in Vim −/− MEFs.** (**a**) Images show GFP and RFP channels and their intensity histograms of WT and Vim −/− MEFs transfected with pMRX-IP-GFP-LC3-RFP-LC3ΔG. Images were taken under steady-state conditions, serum starvation, and nutrient limitation (only EAAs with L-glutamine and glucose in the media) for 3 hours. (**b**) Western blot analysis of WT and Vim −/− MEFs grown for 1 hour in normal DMEM with serum or RPMI media lacking amino acids, glucose, and growth factors supplemented with nutrients in different combinations (1 mM L-glutamine plus 1× EAAs alone or with NEAAs and 2.5 mM glucose, $n$ = 3). The data underlying the histogram shown in the S5A Fig can be found in S1 Data.
(TIF)

**S6 Fig. Overexpression of dnULK1 K46N did not affect cell size in Vim −/− MEFs.** (**a**) Cell volume distribution of Vim −/− MEFs transfected with HAtag-dnULK1 K46N or HA-tag

without insert (*n* = 3). ns = nonsignificant. The data underlying the graphs shown in the S6 Fig can be found in S1 Data.
(TIF)

**S1 Data. Numerical values used to generate the graphs.**
(XLSX)

**S1 Raw images. Uncropped images of the western blots.**
(PDF)

**S1 Table. List of antibodies.**
(PDF)

## Acknowledgments

We would like to thank the Cell Imaging Core at Turku Bioscience and Biocenter Finland for the research facilities.

## Author Contributions

**Conceptualization:** Ponnuswamy Mohanasundaram, John E. Eriksson.

**Funding acquisition:** Franziska Lautenschläger, John E. Eriksson.

**Investigation:** Ponnuswamy Mohanasundaram, Leila S. Coelho-Rato, Mayank Kumar Modi, Marta Urbanska, Fang Cheng.

**Methodology:** Ponnuswamy Mohanasundaram, Leila S. Coelho-Rato, Fang Cheng.

**Supervision:** Franziska Lautenschläger, John E. Eriksson.

**Visualization:** Ponnuswamy Mohanasundaram, Leila S. Coelho-Rato.

**Writing – original draft:** Ponnuswamy Mohanasundaram, Leila S. Coelho-Rato.

**Writing – review & editing:** Ponnuswamy Mohanasundaram, Leila S. Coelho-Rato, John E. Eriksson.

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
