## [Editor Report · Decision Letter 0]

11 Jan 2022

Dear Dr Eriksson, 

Thank you for submitting the revision of your manuscript entitled "Cytoskeletal vimentin regulates cell size and autophagy through mTORC1 signaling" for consideration as a Research Article by PLOS Biology.

We have now had the chance to have a look at the revision and we would like to send it back to the original reviewers. However, as this is a new submission, I am afraid we need you again to provide the metadata that is required for full assessment. To this end, please login to Editorial Manager where you will find the paper in the 'Submissions Needing Revisions' folder on your homepage. Please click 'Revise Submission' from the Action Links and complete all additional questions in the submission questionnaire.

Once your full submission is complete, your paper will undergo a series of checks in preparation for peer review. Once your manuscript has passed the checks it will be sent out for review. To provide the metadata for your submission, please Login to Editorial Manager (https://www.editorialmanager.com/pbiology) within two working days, i.e. by Jan 13 2022 11:59PM.

Given the disruptions resulting from the ongoing COVID-19 pandemic, please expect some delays in the editorial process. We apologise in advance for any inconvenience caused and will do our best to minimize impact as far as possible.

Kind regards,

Ines

--

Ines Alvarez-Garcia, PhD

Senior Editor

PLOS Biology

---

## [Decision Letter · Decision Letter 1]

15 Feb 2022

Dear Dr Eriksson,

Thank you very much for submitting a revised version of your manuscript entitled "Cytoskeletal vimentin regulates cell size and autophagy through mTORC1 signaling" for consideration as a Research Article at PLOS Biology. This revised version of your manuscript has been evaluated by the PLOS Biology editors, the Academic Editor and the three original reviewers.

As you will see, the reviewers appreciate the impressive amount of work that was put in the revision of the manuscript and Reviewer 1 is now fully satisfied. Nevertheless, both Reviewers 2 and 3 would like you to tone down some of the claims to avoid overinterpretation of the results and we do agree with Reviewer 3 on the suggestion of removing glucose from the model (point 4). The reviewers also think that you should strengthen the evidence presented to demonstrate a causal link between suppression of autophagy by vimentin and reduced cell size, and they suggest to measure cell size following inhibition of ULK activity in Vim-/- cells.

In light of the reviews (attached below), we will not be able to accept the current version of the manuscript, but we would welcome re-submission of a revised version that takes into account the reviewers' comments. We cannot make any decision about publication until we have seen the revised manuscript and your response to the reviewers' comments. Your revised manuscript might be sent for further evaluation by the reviewers.

We expect to receive your revised manuscript within 3 months. 

**IMPORTANT - SUBMITTING YOUR REVISION**

3. Resubmission Checklist

a) *PLOS Data Policy*

b) *Published Peer Review*

d) *Blurb*

Please also provide a blurb which (if accepted) will be included in our weekly and monthly Electronic Table of Contents, sent out to readers of PLOS Biology, and may be used to promote your article in social media. The blurb should be about 30-40 words long and is subject to editorial changes. It should, without exaggeration, entice people to read your manuscript. It should not be redundant with the title and should not contain acronyms or abbreviations. For examples, view our author guidelines: https://journals.plos.org/plosbiology/s/revising-your-manuscript#loc-blurb

Sincerely,

Ines

--

Ines Alvarez-Garcia, PhD

Senior Editor

PLOS Biology

Reviewers' comments

Rev. 1:

The authors have nicely addressed the issues raised in the original review.

Rev. 2:

The authors have added new data in the resubmitted paper entitled "Cytoskeletal vimentin regulates cell size and autophagy through mTORC1 signaling", to substantiate their findings. In general, the efforts made by the author to revise their manuscript, especially the mechanism by which vimentin regulates cell size and autophagy, is commendable. However, some issues should be addressed to help to strengthen the impact of this manuscript.

Major points:

1. The authors claimed that "Vim-/- cells showed lower level of phospho-ULK1 as compared to WT upon stimulation and in nutrient depletion conditions", in response to "the authors did not address the point whether suppression of autophagy by vimentin (via activation of mTORC1) is involved in reduced cell size", the 2nd major comment by reviewer 3. However, these data are still observational and do not help to establish a causal link between suppression of autophagy by vimentin and reduced cell size. To address this point, the authors should inhibit ULK activity in Vim-/- cells (e.g., ectopic expression of dominant-negative mutant of ULK) and measure the change in cell size.

2. In figure 5, a parallel western blotting figure is required to asses transfection efficiency of Rag-WT or CA plasmids. Besides, it is better to silence endogenous Rag GTPase before ectopic expression of Rag-WT or CA, which may render the results more significant and convincing.

3. The link between autophagy and cell size should be described briefly either in the introduction or the discussion section.

Rev. 3:

In this revised MS, the authors have made significant efforts in addressing the reviewers' comments. However, to this reviewer, there are several important issues that damps my enthusiasm to this work.

1. In the rebuttal (which is rather lengthy and redundant), the authors failed to provide the following to each of the points raised by the reviewers: (i) is there any new experiment done? (ii) if yes, what types of new experiments and where were the new data presented in the revised MS (either main figures or suppl figures, with figure # and panel #), and (iii) how the text in the MS was revised and where is the revision (page # and line #). Without such information, it is very hard for the reviewers to understand and to assess the revision and improvement made by the authors.

2. Very disappointedly, the authors did not take the reviewers' comments on the proper measurement of autophagy flux seriously. For instance, the authors did not use the lysosomal inhibitors to examine the autophagic flux as clearly suggested by Reviewer 2. The new data added is good but not good enough to reach a solid conclusion on the enhanced autophagy level in Vim-/- cells.

3. The mTORC1 lysosomal localization data presented in Figure 4c and 5a were somehow not consistent and not convincing. For instance, in Figure 4c-top left panel (WT-Starved cells) Lamp1 dots were mainly in the nuclei. In Figure 5a, the two panels at the right, the fluorescence intensity of Lamp1 was much weaker than the two panels at the left. Such inconsistence thus raised significant technical Qs for the validity of the methods used in this study.

4. Throughout the whole study, the effect of glucose on mTORC1 and AKT activation and autopahgy is actually very marginal and minor (Figure 4a, 4b, Figure 6a, 6b, 6d). Therefore, it is not appropriate to include glucose in the illustration in Figure 7.

---

## [Editor Report · Decision Letter 2]

2 Jun 2022

Dear Dr Eriksson,

Thank you for your patience while we considered your revised manuscript entitled "Cytoskeletal vimentin regulates cell size and autophagy through mTORC1 signaling" for publication as a Research Article at PLOS Biology. This revised version of your manuscript has been evaluated by the PLOS Biology editors and the Academic Editor.

Based on ours and the Academic Editor's assessment of your revision, we are likely to accept this manuscript for publication, provided you satisfactorily address the data and other policy-related requests stated below.

We expect to receive your revised manuscript within two weeks. 

*Published Peer Review History*

*Press*

Sincerely,

Ines

--

Ines Alvarez-Garcia, PhD

Senior Editor

PLOS Biology

DATA POLICY:

Thank you for complying with our data policy and providing a data file containing the raw data underlying all the graphs shown in the figures. We have two additional requests:

1) Please indicate in each corresponding figure legend where the underlying data can be found (for example, 'The data underlying the graphs shown in the figure can be found in S1 Data."

2) For figures containing FACS data, we ask that you provide FCS files and a picture showing the successive plots and gates that were applied to the FCS files to generate the figure. If the FCS files are too big, you could deposit them, for example, in the Flow Repository database (Flowcytometry). If you do so, please make the files publicly available and provide the accession number in the Data Availability Statement.

---

## [Editor Report · Decision Letter 3]

1 Jul 2022

Dear Dr Eriksson,

Thank you for the submission of your revised Research Article entitled "Cytoskeletal vimentin regulates cell size and autophagy through mTORC1 signaling" for publication in PLOS Biology. On behalf of my colleagues and the Academic Editor, Alex Gould, I am happy to say that we can in principle accept your manuscript for publication, provided you address any remaining formatting and reporting issues. These will be detailed in an email you should receive within 2-3 business days from our colleagues in the journal operations team; no action is required from you until then. Please note that we will not be able to formally accept your manuscript and schedule it for publication until you have completed any requested changes.

PRESS

Sincerely, 

Ines

--

Ines Alvarez-Garcia, PhD

Senior Editor

PLOS Biology
